# Clinical Features and Outcomes of Enterococcal Bone and Joint Infections and Factors Associated with Treatment Failure over a 13-Year Period in a French Teaching Hospital

**DOI:** 10.3390/microorganisms11051213

**Published:** 2023-05-05

**Authors:** Aurélie Martin, Paul Loubet, Florian Salipante, Paul Laffont-Lozes, Julien Mazet, Jean-Philippe Lavigne, Nicolas Cellier, Albert Sotto, Romaric Larcher

**Affiliations:** 1Infectious and Tropical Diseases Department, Nimes University Hospital, 30000 Nimes, France; aurelie.martin@chu-nimes.fr (A.M.); paul.loubet@chu-nimes.fr (P.L.); paul.laffontlozes@chu-nimes.fr (P.L.-L.); julien.mazet@chu-nimes.fr (J.M.); albert.sotto@chu-nimes.fr (A.S.); 2VBIC (Bacterial Virulence and Chronic Infection), INSERM (French Institute of Health and Medical Research), Montpellier University, 30908 Nimes, France; jean.philippe.lavigne@chu-nimes.fr; 3Department of Biostatistics, Epidemiology, Public Health, and Innovation in Methodology (BESPIM), Nimes University Hospital, 30000 Nimes, France; florian.salipante@chu-nimes.fr; 4Department of Pharmacy, Nimes University Hospital, 30000 Nimes, France; 5Department of Microbiology and Hospital Hygiene, Nimes University Hospital, 30000 Nimes, France; 6Department of Orthopedic surgery and Traumatology, Nimes University Hospital, 30000 Nimes, France; nicolas.cellier@chu-nimes.fr; 7PhyMedExp (Physiology and Experimental Medicine), INSERM (French Institute of Health and Medical Research), CNRS (French National Centre for Scientific Research), University of Montpellier, 34090 Montpellier, France

**Keywords:** prosthetic joint infection, PJI, BJI, orthopedic implant-associated infection, *Enterococcus*, Enterococci, prognosis, biofilm, local inflammation, *Staphylococcus epidermidis*

## Abstract

Enterococcal bone and joint infections (BJIs) are reported to have poor outcomes, but there are conflicting results. This study aimed to describe the clinical characteristics and outcomes of patients with enterococcal BJI and to assess the factors associated with treatment failure. We conducted a retrospective cohort study at Nimes University Hospital from January 2007 to December 2020. The factors associated with treatment failure were assessed using a Cox model. We included 90 consecutive adult patients, 11 with native BJIs, 40 with prosthetic joint infections and 39 with orthopedic implant-associated infections. Two-thirds of patients had local signs of infection, but few (9%) had fever. Most BJIs were caused by *Enterococcus faecalis* (*n* = 82, 91%) and were polymicrobial (*n* = 75, 83%). The treatment failure rate was 39%, and treatment failure was associated with coinfection with *Staphylococcus epidermidis* (adjusted hazard ratio = 3.04, confidence interval at 95% [1.31–7.07], *p* = 0.01) and with the presence of local signs of inflammation at the time of diagnosis (aHR = 2.39, CI 95% [1.22–4.69], *p* = 0.01). Our results confirm the poor prognosis of enterococcal BJIs, prompting clinicians to carefully monitor for local signs of infection and to optimize the medical-surgical management in case of coinfections, especially with *S. epidermidis*.

## 1. Introduction

Among the twelve pathogenic species of *Enterococcus* that have been described, *Enterococcus faecalis* and, to a lesser extent *Enterococcus faecium*, are the two species mostly involved in human infectious diseases [1]. Enterococci are mainly present in the digestive tract of mammals but can also be found in the perineal area in humans, and in the environment, especially in hospitals [2]. They are often associated with other bacteria in polymicrobial infections, contributing to the limited analysis and understanding of their specific virulence and pathogenicity [2]. Enterococci are often considered less virulent than other Gram-positive bacteria pathogenic to humans, such as *Staphylococcus aureus* or *Streptococcus pyogenes* [2]. However, they can cause a wide variety of invasive enterococcal infections, of which the most common clinical manifestations in humans are, in order of frequency, urinary tract infections, intra-abdominal infections, bloodstream infections (including endocarditis), and bone and joint infections (BJIs) [3].

Enterococcal BJI is uncommon and difficult to manage for infectious diseases physicians and orthopedic surgeons and remains a medical-surgical challenge [4,5]. Previous studies [4,5,6,7,8] have mainly reported high treatment failure rates, but conflicting results exist [9]. Consequently, large cohort studies are needed to improve knowledge of the clinical presentation, outcomes, and prognostic factors of enterococcal BJIs.

This study aimed to describe the clinical characteristics and outcomes of patients with enterococcal BJI and to assess the factors associated with treatment failure of native and prosthetic joint and bone infections caused by *Enterococcus* species.

## 2. Materials and Methods

### 2.1. Study Design and Setting

We carried out an observational retrospective cohort study at Nimes University Hospital from January 2007 to December 2020. This 2064-bed teaching hospital has an orthopedic surgery and traumatology department and an infectious diseases department. Clinicians from both departments, pharmacists and microbiologists are involved in the multi-disciplinary team (MDT) meeting for the management of complex BJIs, recognized regional Reference Center for complex Bone and Joint Infections (in French: Centre de Référence des infections Ostéo-Articulaires complexes—CRIOAc—Sud Méditerrannée).

During the MDT meeting that takes place twice monthly, patients with complex BJI are discussed, and their scans and microbiological samples are reviewed to recommend a personalized treatment plan using the combined expertise of team members. The Nimes University Hospital belongs to the French CRIOAc network.

### 2.2. Study Definitions

We defined native BJI according to the French Rheumatology Society guidelines [10].

We defined prosthetic joint infection (PJI) according to the 2020 European Bone and Joint Infections Society (EBJIS) guidelines [11] and the Infectious Diseases Society of America (IDSA) criteria [12].

Three or more cultures that yield the same microorganism were considered definitive evidence of PJI, and a single specimen of a virulent microorganism (such as *S. aureus* or Gram-negative rods) also represented PJI [12].

We classified PJIs according to the time of infection onset after implantation: early (<3 months), delayed (3 months–2 years) and late (>2 years) infections [12].

We defined orthopedic implant-associated infection according to previously published criteria [13].

We classified orthopedic implant-associated infections according to the time of symptoms onset after implantation: early (<3 weeks), delayed (3–10 weeks) and late (>10 weeks) [13].

Treatment failure was defined as the occurrence of at least one of the following events:Relapse: recurrence of the same BJI with the same pathogen at any time after the initiation of the treatment and until the end of follow-up;Reinfection: recurrence of an infection at the same site with the same bacterial species but a different antimicrobial susceptibility testing or with another bacterial species at any time after the initiation of the treatment and until the end of follow-up;Death directly related to BJI.

All diagnoses of native BJI, PJI and implant-associated infections were reviewed by an adjudication committee made up of at least one infectious disease physician, one orthopedic surgeon and one microbiologist. In case of discrepancies, diagnosis was discussed until a consensus was reached among the committee members.

### 2.3. Patients

We screened all samples identified as “*Enterococcus*”, “bone”, “synovial fluid”, “orthopedic surgery”, “joint infection”, and “bone infection” in the microbiology laboratory database during the study period. This sample list was cross-referenced with the CRIOAc database of all patients who had been treated for BJI during the same period. Then, we retrospectively reviewed patient digital charts (Clinicom, InterSystems Corporation, Cambridge, MA, USA) and included all consecutive patients with enterococcal BJI. Patients with diabetic foot osteomyelitis [14], those aged under 18, and pregnant women were excluded.

### 2.4. Microbiological Analyses

Synovial fluids, periprosthetic tissues, and bone biopsies were cultured on Columbia Agar with 5% Sheep blood, Mueller Hinton Chocolate Agar, MacConkey Agar, and Schaedler Broth (bioMérieux, Marcy L’Etoile, France), and incubated at 35 °C ± 2 °C ± in 5% CO_2_ for 14 days.

Between 2009 and 2013, bacterial species were identified with VITEK 2 (bioMérieux) identification cards using colorimetric reading to identify Gram-positive and Gram-negative bacteria (GP and GN cards, respectively). Then, from 2013 to 2015, VITEK-MS (bioMérieux), a Matrix-Assisted Laser Desorption Ionization Time-of-Flight Mass Spectrometry (MALDITOF-MS) System was used for bacterial identification.

The antimicrobial susceptibility was determined using the disk diffusion method on Mueller–Hinton Agar (BioRad, Hercules, CA, USA) according to the European Committee on Antimicrobial Susceptibility Testing (EUCAST) guidelines [15].

### 2.5. Data Collection

We retrospectively collected the following data: age, sex, comorbidities, and tobacco use. We calculated the Charlson comorbidity index for each patient [16]. We recorded the site of BJI and whether an orthopedic implant or a prosthesis was involved. We also noted the occurrence of bacteremia at the time of BJI diagnosis.

We retrospectively collected the microbiological results of bone biopsies and/or synovial fluids obtained during surgery or by aspiration in the operating room and the type of surgery.

### 2.6. Outcomes

After the MDT meeting, patients were followed-up for one year (until 31 December 2021). Outcomes were assessed at 3, 6, and 12 months after the end of the initial antibiotic treatment. Treatment failures requiring additional antimicrobial therapy and/or surgical treatment or causing death were recorded.

### 2.7. Statistical Analysis

The results are expressed as number and percentage for categorical variables and as median (minimum; maximum) or mean and standard deviation (SD) for continuous variables. The population was divided into two groups according to treatment success/failure. Chi-square or Fisher’s exact tests were used to compare categorical variables, and Student’s or Mann–Whitney tests were used on quantitative variables, as appropriate.

We performed survival analyses to consider the temporal dimension. The observation time corresponded to the time between the infection onset and the occurrence of the event (failure) or the end of the observation period (31 December 2021, right-censored) or the date when the patient was lost to follow-up (right-censored).

The factors associated with treatment failure were assessed using Cox models (univariate and multivariate). Based on the results of the univariate Cox model, variables with a *p*-value < 0.1 were retained in the multivariable model. However, variables for which there were too few cases (<3) were discarded, whereas variables such as tobacco consumption, obesity, diabetes, and polymicrobial infection were forced into the model as they are known to influence the prognosis [17,18]. A conditional stepwise regression based on the AIC criterion was performed to select the final model.

Cumulative incidence curves for treatment failure were obtained using the Kaplan–Meier method and compared using the log-rank test.

Statistical analyses were performed with R software version 4.0.3 (The R Foundation for Statistical Computing, Vienna, Austria). All of the tests were two-sided, and a *p*-value of <0.05 was considered statistically significant.

### 2.8. Ethics Approval

The Institutional Review Board of Nimes University Hospital approved the study protocol (IRB number: 22.01.04) and waived the need for signed patient consent. This study was conducted according to the guidelines of the Declaration of Helsinki.

## 3. Results

### 3.1. Population

Ninety patients with enterococcal BJI were analyzed, including 54 males (60%). The median age and Charlson comorbidity index were 69 years (interquartile range (IQR): 54; 82) and 4 (IQR: 2; 4), respectively. The most frequent comorbidities were obesity (*n* = 32, 36%), heart failure (N = 29, 32%) and diabetes (N = 21, 23%). More than one-third of patients were smokers (N = 32, 36%); see Table 1.

At the time of BJI diagnosis, more than half of the patients presented with a purulent discharge (59%) and one-third with local signs of inflammation (37%), but few patients had fever (9%). Clinical signs of infection were mainly found in patients with PJI.

The most frequently involved joints were the hip (N = 26, 29%) and the knee (N = 18, 20%). Forty patients (44%) had a PJI, and 39 (43%) had an orthopedic implant-associated infection, while a native BJI was found in only 11 patients (12%) of the cohort. In patients with PJI or orthopedic implant-associated infection, the mean time between implantation and infection was 2 months (IQR: 1; 23).

Characteristics of patients with enterococcal BJIs are presented in Table 1.

### 3.2. Microbiological Characteristics

The microbiological characteristics of enterococcal BJIs are summarized in Table 2.

The most frequently involved enterococcal species was *Enterococcus faecalis* (*n* = 82, 91%). Seventy-five BJIs were polymicrobial (83%), and 30 (33%) involved at least three pathogens. Coinfections were related to *S. aureus* (38%), *Pseudomonas aeruginosa* (17%), *Proteus mirabilis* (16%), *Enterobacter cloacae* (13%) and *Staphylococcus epidermidis* (11%).

Interestingly, native BJIs (100%) and orthopedic implant-associated infections (92%) were more frequently polymicrobial than PJIs (70%).

### 3.3. Management and Outcomes of Enterococcal BJIs

In most cases, patients with native BJI have undergone surgery, and those with PJI and implant-associated infection had surgical debridement and implant retention (DAIR). All patients were treated with antimicrobials, mainly by oral route, with combination therapy (N = 69, 77%). The most frequently directed therapy used against enterococcal species was amoxicillin (48%), followed by levofloxacin (12%) and linezolid (11%). Patients were treated for 6 or 12 weeks, except for those undergoing a two-step exchange procedure who were treated for a longer time, see Table 3.

Overall, 35 patients with enterococcal BJI had a treatment failure, bringing the failure rate to 39% (Table 3). Treatment failure occurred before the completion of antimicrobial therapy in 30 patients (33%), and most failures were related to device infections (31/35, 89%).

Regarding surgical management of the BJI, all four native BJIs were treated with surgical debridement, 24 of 31 (77%) device-related infections were treated with DAIR, while seven had the device removed (two-step in two patients, one-step exchange procedure in one patient, implant removal in three patients, and arthrodesis in one patient).

Of the 35 patients who failed treatment, 14 (40%) were treated with an anti-enterococcal beta-lactam (13 with amoxicillin and one with piperacillin), six with a glycopeptide, six with linezolid, six with levofloxacin, and rifampicin-based combination therapy was used in 14 cases (40%). All patients received antibiotic treatment as directed by the antimicrobial susceptibility test (Table 4).

In the case of polymicrobial infection, antibiotic treatments were directed according to the available antimicrobial susceptibility tests.

### 3.4. Factors Associated with Enterococcal BJI Treatment Failure

In univariate analysis, the factors associated with treatment failure were local inflammation (hazard ratio (HR) = 2.17, 95% confidence interval (95% CI) [1.11–4.21], *p* = 0.02), coinfection with coagulase-negative staphylococci (HR = 2.32, 95% CI [1.05–5.12], *p* = 0.04) and particularly coinfection with *S. epidermidis* (HR = 2.59, 95% CI [1.13–5.95], *p* = 0.02). No specific medical or surgical management was associated with treatment failure in the univariate analysis.

In multivariable analysis, only local inflammation (adjusted hazard ratio (aHR) = 2.39, 95% CI [1.22–4.69], *p* = 0.01) and coinfection with *S. epidermidis* (aHR = 3.04, 95% CI [1.31–7.07], *p* = 0.01) remained associated with treatment failure, see Table 5.

Cumulative incidence curves of treatment failure in patients with and without local inflammation at the time of diagnosis of BJI and in patients with and without *Staphylococcus epidermidis* coinfections are depicted in Figure 1.

Patients with polymicrobial infections tended to have higher failure rate than those with monomicrobial infections, 41%, CI 95% [30–53%] versus 27%, CI 95% [9–55%], respectively, but this difference was not statistically significant (*p* = 0.44).

Most interestingly, focusing on the 79 patients with orthopedic devices, we also highlighted that the rate of treatment failure tended to be higher in those who had DAIR than in those who had an implant removal (*p* = 0.1), see Appendix A.

## 4. Discussion

In this study, we investigated a thirteen-year French cohort of 90 patients with enterococcal BJIs, which were mainly caused by *E. faecalis* and reported a failure rate of 39%. More than half of the patients presented with a purulent discharge and more than one-third presented with local signs of inflammation at the time of diagnosis of BJIs. Most patients had polymicrobial BJIs, and we found that coinfection with *S. epidermidis* and the presence of local signs of inflammation were independently associated with therapeutic failure. Finally, among patients with PJIs or orthopedic implant-associated infections, management with DAIR tended to be associated with worse outcomes.

In accordance with previous reports [5], we found that BJIs caused by *Enterococcus* sp. were difficult-to-treat infections associated with a high treatment failure rate. Our results were in line with previous studies [4,6,7,8], which reported failure rates ranging from 28% to 48%, with the exception of a multicenter cohort from Germany which reported a lower failure rate of 16% in 77 patients with PJIs [9]. However, in this study [9], reinfections were not considered as treatment failures, in contrast to other studies [4,6,7,8]. Several factors known to worsen the prognosis of BJIs, such as obesity, diabetes mellitus and tobacco consumption [17,18], were found in high proportion in patients of our cohort and others [4,6,7,8], which may explain the high failure rates. In addition, some authors [8] have suggested that the ability of enterococci to form biofilm may inhibit antimicrobial activity and contribute to a poorer prognosis in enterococcal BJIs.

Polymicrobial infections are common in patients with enterococcal BJIs [4,6,7,8] and are known to be another risk factor involved with treatment failure [17,18]. Accordingly, we found that patients with polymicrobial infections had poorer outcomes than those with monomicrobial infections. Particularly, we found that coinfection with *S. epidermidis* was an independent risk factor for the treatment failure of enterococcal BJIs. The ability of *S. epidermidis* to form a biofilm, especially in patients with PJIs, is reputed to increase antibiotic resistance and treatment failure [19,20]. We can hypothesize that *S. epidermidis* is able to cooperate within the biofilm with different species of enterococci, notably through the quorum-sensing mechanism [21]. Quorum-sensing allows individual bacteria within colonies to coordinate and carry out colony-wide functions, such as virulence or biofilm formation, which plays a major role in the resistance to antimicrobials in enterococci [21]. The Fsr quorum-sensing system in *E. faecalis* regulated gelatinase and serine protease, which are involved in virulence, host tissue degradation and biofilm formation [21]. Thus, the development of new drugs with quorum-sensing inhibitory action could be a therapeutic option to improve the prognosis of these patients in the future [21].

While awaiting new antimicrobial therapeutic innovations, some authors have suggested the systematic use of antibiotics effective against biofilm [7]. Fluoroquinolones, in particular, levofloxacin and moxifloxacin, which have the lowest minimum inhibitory concentrations for enterococcal species, have been proposed, but data on their use for the treatment of enterococcal infections are scarce [22,23]. Rifampicin-based combination therapy has been reported to be associated with better outcomes [8]; however, our study did not confirm this result (Appendix A). Alternative treatment options may include Fosfomycin [9] or a combination of ampicillin and ceftriaxone [24], but the best antimicrobial regimen remains debated and unknown [25]. No antimicrobial was associated with therapeutic success or failure in our study; nonetheless, our results suggest the importance of optimized surgical management, including the removal of prostheses and orthopedic implants where possible, especially in delayed infections, as previously reported [8].

A striking finding of our study was that local inflammation at the time of diagnosis of BJIs was associated with poorer outcomes. At the same time, we reported that purulent discharge and local pain were common, whereas fever was rare in patients with enterococcal BJIs. Some authors [25] reported that enterococci were similar to *Cutibacterium acnes* or coagulase-negative staphylococci in terms of virulence and that the development of clinical symptoms of BJIs was slow [25], which was in agreement with the long time to infection reported in our study (median 2 months). This may indicate that signs of locoregional spread of BJIs are indicative of an advanced infection and, therefore, an increased risk of treatment failure. These findings could prompt physicians to carefully monitor local signs in patients with suspected PJI or implant-associated infection.

This study has limitations, including its observational nature, with a possible heterogeneity of the study population that includes native joint infections, joint prostheses and osteosynthesis hardwires. However, this cohort study allowed the evaluation of a homogeneous management strategy by the same team. Furthermore, the study’s retrospective design, over a period of 13 years, did not allow us to perform a whole-genome multilocus sequence typing (wgMLST) to characterize the strains further, and misidentification may have impacted the prognosis. Second, the retrospective and single-center design could also limit the generalization of the results. Nevertheless, the rates of therapeutic failure that we reported are consistent with those previously reported [4,6,7,8]. Third, as in all studies based on BJI/PJI definitions, some bacteria considered as contamination may not have been treated and may have a pathogenic role and modify the prognosis. Finally, the small cohort size could limit the detection of group differences in management and outcomes. Nonetheless, to our knowledge, this cohort of enterococcal BJIs is the largest reported in the literature.

## 5. Conclusions

In this large cohort of enterococcal BJIs, including native BJIs, PJIs, and implant-associated infections, patients presented mainly with local signs of infection and rarely with fever. *E. faecalis* was the most frequently isolated enterococcal species, followed by *E. faecium*, and in most cases, BJIs were polymicrobial. The treatment failure rate was high, estimated at 39%, and treatment failure was independently associated with coinfection with *S. epidermidis* and with local signs of inflammation at the time of diagnosis of BJIs.

Further studies are mandatory to confirm our results and to assess the best antimicrobial and surgical strategies to improve the outcomes of patients with enterococcal BJIs. In particular, the role of surgery and its timing, and the role of combination therapies of ampicillin and ceftriaxone, or combination with fosfomycin or rifampicin, need to be prospectively evaluated to determine the management that could improve the prognosis of enterococcal BJIs.

## Figures and Tables

**Figure 1 microorganisms-11-01213-f001:**
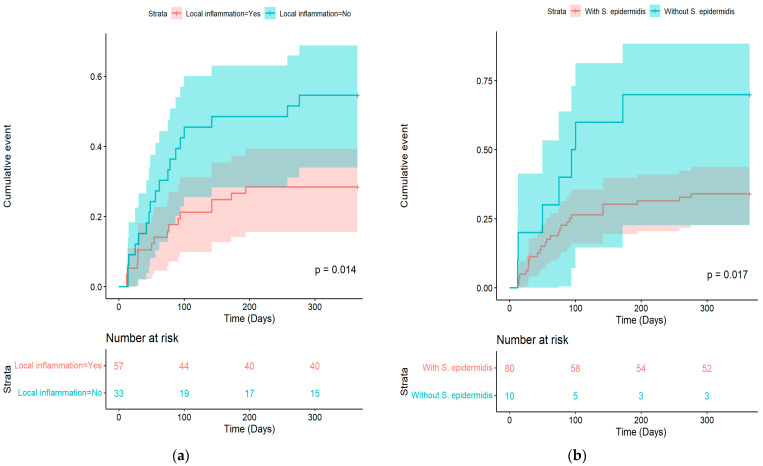
(**a**) Cumulative incidence curves of treatment failure in patients with and without local inflammation at the time of diagnosis and (**b**) Cumulative incidence curves of treatment failure in patients with and without *Staphylococcus epidermidis* coinfections.

**Table 1 microorganisms-11-01213-t001:** Characteristics of patients with enterococcal bone and joint infections (BJIs).

Characteristics (N = 90)	N (%) or Median (IQR)
Age	69 (54; 82)
Sex, male	54 (60%)
Comorbidities ^1^	
Chronic kidney disease	11 (12%)
Liver failure	2 (2%)
Heart failure	29 (32%)
Immunodepression	4 (4%)
Diabetes	21 (23%)
Cirrhosis	3 (3%)
Cancer chemotherapy	3 (3%)
Obesity ^2^	32 (36%)
HIV ^3^ or hepatitis	6 (6%)
Tobacco consumption	32 (36%)
Charlson comorbidity index	4 (2; 4)
Clinical features:	
Purulent discharge	53 (59%)
Local inflammation	33 (37%)
Local pain	23 (26%)
Fever (≥38.5 °C)	8 (9%)
Bacteremia/septic metastasis	2 (2%)
Bones and joints involved:	
Hip	26 (29%)
Knee	18 (20%)
Shin	13 (14%)
Ankle	11 (12%)
Foot	13 (14%)
Others ^4^	8 (9%)
Native BJI	11 (12%)
Implant-associated infection	39 (43%)
Prosthetic joint infection (PJI)	40 (44%)
Infection delay (months)	2 (1; 23)
Early onset	24 (27%)
Delayed onset	25 (28%)
Late onset	30 (33%)

^1^ Patients could have more than one comorbidity. ^2^ Body Mass Index > 30 kg/m^2^. ^3^ Human immunodeficiency virus; ^4^ Thigh (N = 7), elbow (N = 1), hand (N = 1).

**Table 2 microorganisms-11-01213-t002:** Microbiological characteristics of enterococcal bone and joint infections (BJIs).

Characteristics	Overall (N = 90) N (%)	Native BJI (N = 11) N (%)	Implant-Associated Infection (N = 39) N (%)	PJI (N = 40) N (%)
Enterococcal species				
*Enterococcus faecalis*	82 (91%)	11 (100%)	34 (87%)	37 (93%)
*Enterococcus faecium*	8 (9%)	0 (0%)	5 (13%)	3 (7%)
*Enterococcus gallinarum*	2 (2%)	0 (0%)	1 (3%)	1 (3%)
*Enterococcus avium*	1 (1%)	0 (0%)	1 (3%)	0 (0%)
*Enterococcus casseliflavus*	1 (1%)	0 (0%)	1 (3%)	0 (0%)
Polymicrobial infections ^1^	75 (83%)	11 (100%)	36 (92%)	28 (70%)
*Staphylococcus aureus*	34 (38%)	2 (18%)	19 (49%)	13 (33%)
*Staphylococcus epidermidis*	10 (11%)	0 (0%)	6 (15%)	4 (10%)
*Staphylococcus lugdunensis*	5 (6%)	1 (9%)	2 (5%)	2 (5%)
*Streptococcus* sp.	8 (9%)	3 (27%)	4 (10%)	1 (3%)
*Corynebacterium* sp.	11 (12%)	3 (27%)	5 (13%)	3 (8%)
Enterobacterales	40 (44%)	9 (82%)	15 (38%)	18 (45%)
*Pseudomonas aeruginosa*	15 (17%)	2 (18%)	7 (18%)	6 (15%)
*Finelgodia magna*	6 (7%)	1 (9%)	4 (10%)	1 (3%)
*Propionibacterium acnes*	2 (2%)	0 (0%)	1 (3%)	1 (3%)
*Peptostreptococcus anaerobius*	1 (1%)	0 (0%)	0 (0%)	1 (3%)
Other anaerobes	11 (12%)	6 (55%)	2 (5%)	3 (8%)

^1^ Patients could have more than one coinfectant pathogen.

**Table 3 microorganisms-11-01213-t003:** Enterococcal bone and joint infections (BJIs) management and outcomes.

Characteristics	Overall (N = 90) N (%)
Antimicrobial therapy:	
Duration of 6 weeks	36 (40%)
Duration of 12 weeks	50 (56%)
Duration > 12 weeks	4 (4%)
Oral route	76 (84%)
Amoxicillin	43 (48%)
Levofloxacin	11 (12%)
Linezolid	10 (11%)
Piperacillin	8 (9%)
Vancomycin	6 (7%)
Teicoplanin	5 (6%)
Imipenem	3 (3%)
Daptomycin	1 (1%)
Rifampicin combination therapy	37 (41%)
Non-anti-enterococcal antibiotics *	38
Surgical management:	
Debridement of native bone/joint	10 (11%)
Debridement and implant retention (DAIR)	52 (58%)
1-step exchange procedure	1 (1%)
2-step exchange procedure	4 (4%)
Implant removal	16 (18%)
Arthrodesis	1 (1%)
Amputation	4 (4%)
None	2 (2%)
Outcomes:	
Treatment failure	35 (39%)
Relapse	32 (36%)
Reinfection	1 (1%)
Death	2 (2%)

* ofloxacin (N = 27), ciprofloxacin (N = 8), cotrimoxazole (N = 5), ceftriaxone/cefotaxime (N = 4), fusidic acid (N = 3), clindamycin (N = 2), cefepime (N = 1), ertapenem (N = 1).

**Table 4 microorganisms-11-01213-t004:** Antimicrobial susceptibility of enterococcal species involved in bone and joint infections.

Enterococcal Species	Percentage of Susceptible Strain									
	AMP	AMX	PIP	IPM	VAN	TEC	LZD	LVX	SXT	RIF
*E. faecalis* (N = 82)	100%	100%	100%	100%	100%	100%	100% ^1^	89% ^1^	53% ^2^	79% ^3^
*E. faecium* (N = 8)	63%	63%	40% ^4^	40% ^4^	100%	100%	100%	100% ^5^	50%	0%
*E. gallinarum* (N = 2)	100%	100%	0%	100%	0%	100%	100%	100% ^6^	50%	0% ^6^
*E. avium* (N = 1)	100%	100%	0%	100%	100%	100%	100%	-	100%	100%
*E. casseliflavus* (N = 1)	100%	100%	0%	100%	0%	100%	100%	-	0%	0%

^1^ of 19 strains; ^2^ of 80 strains; ^3^ of 14 strains; ^4^ of 5 strains; ^5^ of 2 strains; ^6^ of 1 strain. Abbreviations: AMP: ampicillin; AMX: amoxicillin; PIP: piperacillin; IPM: imipenem; VAN: vancomycin; TEC: teicoplanin; LZD: linezolid; LVX: levofloxacin; SXT: trimethoprim–sulfamethoxazole; RIF: rifampin.

**Table 5 microorganisms-11-01213-t005:** Factors independently associated with enterococcal BJI treatment failure.

Characteristics (N = 90)	Success N (%)	Failure N (%)	Univariate Analysis			Multivariable Analysis		
			HR	95% CI	*p*-Value	aHR	95% CI	*p*-Value
Local inflammation	15 (45.5%)	18 (54.5%)	2.32	(1.05; 5.12)	0.02	2.39	(1.22; 4.69)	0.01
SCN * coinfections	4 (33.3%)	8 (66.7%)	2.32	(1.05; 5.12)	0.04			
*S. epidermidis* coinfections	3 (30%)	7 (70%)	2.59	(1.13; 5.95)	0.02	3.04	(1.31; 7.07)	0.01

* Coagulase-negative Staphylococci. HR: hazard ratio, aHR: adjusted hazard ratio, 95% CI: confidence interval at 95%.

## Data Availability

The authors consent to share the data collected with others. The raw data supporting the conclusions of this article will be made available by the authors without undue reservation. The data will be available immediately after the main publication and indefinitely.

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
