# Peer review of "Clinical Features and Outcomes of Enterococcal Bone and Joint Infections and Factors Associated with Treatment Failure over a 13-Year Period in a French Teaching Hospital"

_microorganisms, 2023, doi:10.3390/microorganisms11051213_

Round 1
Reviewer 1 Report
The overall logic of the article is clear and the topic selection is novel, but there are still some minor flaws.

The full text has a smooth word order and no spelling errors。
Author Response
The manuscript is entitled “Clinical features and outcomes of enterococcal bone and joint infections and factors associated with treatment failure over a 13-year period in a French teaching hospital”.
In this work, the authors was to describe the clinical characteristics and outcomes of patients with enterococcal BJI and to assess the factors associated with treatment failure of native and prosthetic joint and bone infections caused by Enterococcus species. It can confirm the poor prognosis of enterococcal BJIs, prompting clinicians to carefully monitor for local signs of infection and to optimize the medical-surgical management in case of coinfections, especially with S. epidermidis. However, we think there are still some issues in this manuscript need to be corrected by the author before publication.
First, we thank the reviewer for their interest in this work and thorough comments.
(1)Some characteristics are not in proportion to those described in Table 1.
Thanks to the reviewer for highlighting this mistake. The text has been revised. Please see Table 1 and text p4.
(2)The description of the treatment and outcome of the disease in Table 3 is too little, and detailed explanations are recommended.
As requested by the reviewer, we have added details about the treatment and outcomes. Please see p7.
(3)No specific improvements are proposed to the limitations studied in this paper.
The conclusion has been revised according to the reviewer remark. Please see the conclusion section.
(4)Sensitivity to microbial antimicrobials is not described.
We agree with the reviewer that antimicrobial susceptibility testing (AST) results are important data to report in this paper. We thank the reviewer for his helpful comments. We have added a table in the article to inform about antimicrobial susceptibility testing. Please see Table 4 p7.
Reviewer 2 Report
Manuscript Martin et al., Clinical features and outcomes of enterococcal bone and joint infections and factors associated with treatment failure over a 13-year period in a French teaching hospital.
Infections caused by Enterococcus represent a significant problem and it is commendable that the authors focused on this problem. Even in this study, the authors confirmed up to 39% of treatment failure caused by Enterococci infection. The manuscript represents a well-designed statistical analysis of infections in 90 patients over 13 years. The group of patients is balanced and several comorbidities, such as chronic kidney disease, obesity, liver failure, or diabetes, were taken into account in the design of the study. Despite of these advantages, I have few comments on the manuscript.
Major:
1. In the manuscript, you state that the isolates were identified by two methods (VITEK2 and VITEK-MS). It would be appropriate to mention in the discussion the possible impact on species determination, especially in the case of polymicrobial infections.
2. When selecting data for the study, you excluded patients with diabetic foot osteomyelitis or pregnant women from the cohort. Given the nature of the study, it is not clear to me why. Were they patients with BJI or PJI? Please state the reason in the discussion.
3. In the study, you did not report the sensitivity of the isolates to antibiotics, which significantly affect the overall course of the infection. Also, it is not clear from the work whether a closer genetic analysis was also done (MLST, 16S rRNA, AFLP...). Is it possible that these were also nosocomial strains? Is it possible to provide such data?
Minor:
1. The work is well written, but the authors did not avoid difficult-to-understand sentences (especially lines 187-189)
2. In Table 1, do you indicate the age of the entire group or only the men?
3. Table 2 shows a total of two cases of Enterococcus gallinarum infection, but only one case of implant-associated infection is listed. To which group does the missing one belong?
4. Please provide a list of abbreviations, for example the abbreviation PJI (prosthetic joint infections) is not explained in the text.
Author Response
Manuscript Martin et al., Clinical features and outcomes of enterococcal bone and joint infections and factors associated with treatment failure over a 13-year period in a French teaching hospital.
Infections caused by Enterococcus represent a significant problem and it is commendable that the authors focused on this problem. Even in this study, the authors confirmed up to 39% of treatment failure caused by Enterococci infection. The manuscript represents a well-designed statistical analysis of infections in 90 patients over 13 years. The group of patients is balanced and several comorbidities, such as chronic kidney disease, obesity, liver failure, or diabetes, were taken into account in the design of the study. Despite of these advantages, I have few comments on the manuscript.
The authors thank the reviewer for their interest in our work and their remarks.
Major:
- In the manuscript, you state that the isolates were identified by two methods (VITEK2 and VITEK-MS). It would be appropriate to mention in the discussion the possible impact on species determination, especially in the case of polymicrobial infections.
We agree with the reviewer that species determination may impact the prognosis of infections. In this 13-year retrospective study, most strains were no longer available in the laboratory to perform a wgMLST. We have acknowledged this limitation in the dedicated section of the discussion.
- When selecting data for the study, you excluded patients with diabetic foot osteomyelitis or pregnant women from the cohort. Given the nature of the study, it is not clear to me why. Were they patients with BJI or PJI? Please state the reason in the discussion.
We thank the reviewer for this careful approach to selection bias. Foot infections in diabetic patients are known to have a different pathophysiology (with different grades and severity), management, and prognosis than other bone and joint infections (see Lipsky et al. CID 2012, https://doi.org/10.1093/cid/cis346). They are usually analyzed separately because they have a different nosology and microbial ecology, and we felt it necessary to exclude these patients from our cohort.
Regarding pregnant women, they are often excluded from studies (even observational and retrospective) at the request of our IRB. However, no pregnant women had BJI/PJI during the study period.
- In the study, you did not report the sensitivity of the isolates to antibiotics, which significantly affect the overall course of the infection. Also, it is not clear from the work whether a closer genetic analysis was also done (MLST, 16S rRNA, AFLP...). Is it possible that these were also nosocomial strains? Is it possible to provide such data?
As correctly pointed out by the reviewer, we did not report the sensitivity of the isolates to antibiotics, and we completely agree with the reviewer that it could significantly affect the overall course of the infection. Thanks to the reviewer for highlighting this helpful observation. We have added a table in the article to inform about antimicrobial susceptibility testing. Please see Table 4 p7.
We could not perform wgMLST analysis in this study and have acknowledged this limitation in the dedicated section of the discussion. However, we do not believe these are nosocomial strains given the distribution in time (13 years study) and space (1 infectious diseases department, 2 orthopedic departments and several operating rooms in our university hospital).
Minor:
- The work is well written, but the authors did not avoid difficult-to-understand sentences (especially lines 187-189)
Thanks to the reviewer for these remarks. We revised the manuscript in accordance with the reviewer’s comments.
- In Table 1, do you indicate the age of the entire group or only the men?
We indicated the age of the entire group. We have changed the text to be more explicit. Please see Table 1.
- Table 2 shows a total of two cases of Enterococcus gallinarum infection, but only one case of implant-associated infection is listed. To which group does the missing one belong?
Thanks to the reviewer for highlighting this mistake. There is two BJI caused by Enterococcus gallinarum in our cohort, one implant-associated infection and one prosthetic joint infection. Table 2 has been modified accordingly.
- Please provide a list of abbreviations, for example the abbreviation PJI (prosthetic joint infections) is not explained in the text.
We thank the reviewer again for highlighting this mistake. The “Information for Authors” of Microorganisms journal asks authors to define the abbreviation the first time they use it in the text (or in a table/figure), not to provide a list of abbreviations. According to the reviewer’s observation and the Information for Authors, we define PJI as prosthetic joint infections. Please see text p2.